# A national survey on current clinical practice pattern of Korean Medicine doctors for treating obesity

Kyung Hwan Jegal[1,2], Mi Mi Ko[3], Bo-Young Kim[3], Mi Ju Son[3], Sungha Kim[3]*

**1** Digital Health Research Division, Korea Institute of Oriental Medicine, Daejeon, Republic of Korea,
**2** College of Korean Medicine, Daegu Haany University, Gyeongsan, Republic of Korea, **3** KM Science Research Division, Korea Institute of Oriental Medicine, Daejeon, Republic of Korea

* bozzol@kiom.re.kr

## Abstract

### Background and aims

Given the multifactorial nature of obesity, there is current interest on Korean medicine (KM) for weight loss. This survey aimed to investigate current practice patterns of KM treatment for obesity among doctors.

### Methods

A questionnaire on clinical practice patterns of KM treatment for obesity was constructed and distributed to 21,788 KM doctors (KMDs). The questionnaire was consisted of respondent characteristics, state of treated patient, diagnosis, treatment, and usage pattern of herbal medicine for obesity.

### Results

A total of 4.9% of KMDs (n = 1,084/21,788) validly completed the survey. Patients with Obesity Class I ($25 \leq$ Body mass index (BMI) $\leq 29.9$) are the most common in KM clinics. Bioelectric impedance and KM Obesity Pattern Identification Questionnaire were routinely used for diagnosis. The average treatment duration was 4.16 weeks, and patients visited on an average 4.25 times per month for treatment. Herbal medicine is the most commonly used for obesity treatment by KMDs, and Taeeumjowui-tang was the most frequently prescribed. Ephedrae Herba, which is identified as the most used herbs for weight loss, was prescribed 5.18 ± 2.7 g/day at minimum and 10.06 ± 4.23 g/day at maximum. A total of 62.9% of responded KMDs had ever a patient with uncomfortable symptoms due to Ephedrae Herba use, neuropsychiatric events were the most common symptoms, followed by gastrointestinal events.

### Conclusion

Taken together, this study provides information on real clinical practice patterns of KM including patients, diagnosis, treatments, and herbal medicine for obesity.

**Data Availability Statement:** All relevant data are within the paper and its Supporting Information files.

**Funding:** This research was supported by a grant of the Korea Health Technology R&D Project through the Korea Health Industry Development Institute (KHIDI), funded by the Ministry of Health & Welfare, Republic of Korea (Grant No.: HF20C0208). The funders had no role in study design, data collection and analysis, decision to publish, or preparation of the manuscript.

**Competing interests:** The authors have declared that no competing interests exist.

**Abbreviations:** BMI, Body mass index; FDA, the US Food and Drug Administration; KM, Korean Medicine; KMD, Korean Medicine Doctor; KNHNES, the Korea National Health and Nutrition Examination Survey.

## Introduction

Obesity is a major risk factor for various diseases, such as cancer, cardiovascular diseases and metabolic diseases. The incidence of obesity and overweight has been increasing gradually over the past decades, and it is becoming a public health problem. In South Korea, the prevalence of obesity continuously increased in the last decade, from 29.7% in 2009 to 35.7% in 2018 (45.4% in males and 26.5% in females, respectively) in the total population [1]. Moreover, it is predicted to increase steadily and reach 62% in males and 37% in females by 2030 [2]. The burden of social and economic costs has been increasing immensely. The total social and economic losses for obesity in Korea was 11.5 trillion KRW (10 billion USD) in 2016, which represents 0.7% of the Korean gross domestic product, and approximately 50% of the costs were medical expenses [3].

According to the Korean National Health and Nutrition Examination Survey (KNHNES) data, as the obese population increased, attempts to lose weight also continuously increased, from 18% in males and 31% in females in 2001 to 36% and 48% in 2014, respectively [4]. The first line treatment for obesity is lifestyle interventions such as diet control, physical activity, and behavior therapy. If a first line treatment for obesity is unsuccessful, medication and bariatric surgery are considered as secondary therapeutic options [5]. Although bariatric surgery has been covered by the National Health Insurance since 2019 in Korea, it is only allowed for morbidly obese patients (BMI $\geq$ 35 kg/m$^2$ or BMI $\geq$ 30 kg/m$^2$ with comorbidities, such as hypertension and diabetes). In addition, only two anti-obesity drugs, orlistat and lorcaserin, are approved for long-term use in Korea; however, serious side effects, such as liver injury, acute kidney injury, pancreatitis, and cardiac valvulopathy, remain [6]. KM is recognized as the double axis of the Korean health-care system along with conventional Western medicine. Supportive evidence has also been accumulated on the safety and effectiveness of herbal medicine for obesity compared with conventional medicines, placebos, or lifestyle control [7, 8]. Moreover, acupuncture is an effective intervention for obesity and its anti-obesity effects may be maximized when combined with lifestyle control [9, 10]. Furthermore, herbal medicine is one of the most effective strategy in the Korean population according to the KNHNES [11, 12], and has shown significant anti-obesity effects in randomized controlled trials, such as Bangpungtongseong-san [13, 14], Euiiyin-tang [15], and Taeeumjowui-tang [16]. Most of the herbal formulas that show anti-obesity effects are considered to contain Ephedrae Herba, one of the most commonly used medicinal herbs for weight loss in recent years. This herb contains ephedrine, a phenylpropylamine protoalkaloid derived from Ephedra species, which is reported to enhance energy expenditure, lipid and glucose metabolism, and promote weight loss by stimulating thermogenesis [17]. However, ephedra and ephedrine are associated with several adverse drug events, such as hypertension, palpitation, tachycardia, insomnia, and gastrointestinal side effects [18]. In response to accumulating evidences of its safety problems, in 2004, the US Food and Drug Administration (FDA) prohibited the sales of dietary supplement containing ephedrine alkaloids (ephedra). Ephedrae Herba is also prohibited by the Korean Ministry of Food and Drug Safety for food or food additives, but only prescribed by licensed Korean Medicine (KM) doctors (KMDs) for medical use to treat cold, flu, fever, chill, cough, nasal congestion, bronchial asthma and obesity [19]. However, Korean National Health Insurance has low coverage for KM obesity treatments, including prescribed herbal medicine. Even basic information about these non-insured treatments for obesity has not been well documented, such as the number of patients, treatment duration, and prescribed medication and its adverse events. Thus, information or statistic data about real clinical practice field of KM for obesity treatment are scarce.

In the present study, we conducted an online survey targeting KMDs to obtain information about the current status on clinical practice including patients, diagnosis, treatments and herbal medicine for obesity treatment.

## Methods

### Participants and recruitment

This study is a survey study conducted to investigate the current clinical practice pattern of obesity treatment conducted by KMDs. All licensed KMDs are registered members of the Association of Korean Medicine, and a total of 21,788 KMDs were surveyed via e-mail using the Survey Monkey website (https://www.surveymonkey.com). This survey was conducted from February 4 to March 8, 2021. The e-mail enclosed URL of the QR code linked to the web-based questionnaire and a cover letter explaining the purpose of the survey and use of obtained data. The link was opened for 33 days from the beginning of the survey, and reminders were sent to increase the response rate. The survey was conducted anonymously, and the data were downloaded from the Survey Monkey website without any personal identifying information.

### Development of survey questionnaire

The questionnaire is a self-report web-based survey, which has been developed to elicit qualitative and quantitative information from the actual clinical practice field of KM for obesity. The initial draft was constructed by KMD researchers with reference to the Korean Medicine Clinical Practice Guideline for obesity [20]. Subsequently, it was revised by referring to comments from KMDs who are involved in obesity treatment. The questionnaire consisted of 38 questions, including 7 sub-questions.

For demographic data, questions on the general characteristics of responders, such as their gender, age, clinical experience, education, place of work, and KM specialists, were included in the first section. In particular, the question about the place of work is designed to identify whether it is a clinic specialized in obesity treatment, and subgroup analysis was conducted based on the response to this question. The second section consisted of questions about the current clinical status, such as the annual number of patients, major sex, age, obesity level, treatment duration, and frequency of visits of the patients receiving care. The classification of obesity level by BMI is represented in the 2018 Korean Society for Study of Obesity Guideline for the Management of Obesity in Korea, which is in accordance with the WHO obesity guideline for the Asia-Pacific region [5]. The questionnaires also investigated diagnostic tools and interventions used in obesity care, as well the average degree of weight loss after treatment. The next section involved questions about the specific usage patterns of herbal medicine and status of post-treatment management for obesity. In particular, the questions, such as the prescription dose of Ephedrae Herba and its determinants, were included to determine the pattern of Ephedrae Herba use for obesity and its related discomfort symptoms complained by patients. The estimated time to complete the survey was within 10 min. Participants who did not complete the survey or responded to the survey multiple times were excluded. In order to prevent missing data, the survey was designed such that it will not be considered completed if the participants intentionally skipped any question. In case the intent of the answer by free text format was unclear due to obscure terminology, it was reported to KMD researchers to reduce errors in data analysis. If a questionnaire was forcibly terminated in the middle of the survey, it was excluded from the analysis.

### Ethical considerations

The present study was approved by the Institutional Review Board (IRB, No. I-2011/009-003) of the Korea Institute of Oriental Medicine. All participants were provided with sufficient explanations on the purpose and contents of the study and voluntarily participated in the survey. Informed consent about protection of personal information and the use of collected data for academic purposes was obtained from all participants.

## Statistical analysis

Descriptive statistics are presented as frequency and percentage distributions for categorical data, and continuous variables are presented as mean ± standard deviation. Significant differences for each outcome between the groups were evaluated using independent two sample t-tests or Wilcoxon rank sum tests for continuous variables and chi-square tests or Fisher's exact tests for categorical variables. The data were analyzed using SAS software, version 9.4 (SAS Institute Inc., Cary, NC, USA), and the level of significance was set to 0.05, and two-tailed comparisons were performed.

## Results

### Participant characteristics

Only 1695 of 21,788 KMDs who received the e-mail responded to the survey. Among them, 139 respondents were removed due to multiple responses, in addition those who terminated the questionnaire in a middle of the survey. A total of 1084 survey responses were analyzed. Demographic characteristics are represented in Table 1. As for the age, 30s aged respondents (n = 515) accounted for the largest share at 47.5%. KMDs with over 10 years of clinical experience comprised 40.8% (n = 442) of all respondents, while those with less than 10 years of experience comprised 31.5%. KM specialist doctors comprised 23.25% (n = 252) of the respondents. Only 51 respondents answered that they were working at an KM clinic specialized in obesity treatment., and the remaining were classified as working at non-specialized clinics/hospitals for further sub-group analysis.

### Current status of KM clinical practice in obesity treatment

Table 2 shows the current information about obesity clinical practice in KM. The average period of clinical experience was 6.70 years in obesity treatment. The average number of obese patients treated by KMDs was 107.13 per year. In contrast, respondents from obesity-specialized clinics treat an average of 1129.47 patients per year, which is significantly more than KMDs from non-specialized clinics who treat 56.65 patients per year ($p < .0001$). Most patients receiving KM treatment were females, and the main age groups were 30s or 40s. Most respondents (n = 469, 43.3%) reported that the average obesity of the patients treated was the Obesity Class I (25 ≤ Body mass index (BMI) ≤ 29.9), followed by the Obesity Class II (BMI 30 ~ 34.9) and overweight patients (23 ≤ BMI ≤ 24.9). No significant difference was noted in the distribution according to obesity treatment specialization of clinics. Metabolic syndrome was most frequently reported as the primary complication of obese patients (n = 686, 63.3%). The average treatment duration was 4.16 weeks, and patients visited average 4.25 times a month for treatment. The treatment duration of 10–20 minutes per visit was the most common among KMDs (n = 467, 43.1%). In addition, the average treatment duration in the obesity-specialized KM clinics (4.53 weeks) was significantly longer than that of the non-specialized clinics/hospitals (4.14 weeks) ($p < 0.0001$). The average weight loss after treatment was 7.15 kg and loss of 10.56% of body weight was noted in the total group analysis. Furthermore, significantly more weight loss was noted in specialized clinics (average, 12.22% of body weight) than in non-specialized clinics/hospitals (average, 10.48% of body weight).

### Diagnostic tools and therapeutic intervention in KM for obesity treatment

The results of survey on the methods of diagnosis, and therapeutic intervention and its determinants used in KM treatment for obesity are represented in Table 3. The most used diagnostic tool or device used was bioelectrical impedance (n = 925, 85.3%), and 43.6% of respondents

**Table 1. Demographic characteristics of respondents (n = 1084).**

| Classification | | N (%) |
|---|---|---|
| Gender | | |
| | Male | 693 (63.9) |
| | Female | 391 (36.1) |
| Age(years) | | |
| | ≤ 29 | 199 (18.4) |
| | 30–39 | 515 (47.5) |
| | 40–49 | 261 (24.1) |
| | 50–59 | 94 (8.7) |
| | ≥ 60 | 15 (1.4) |
| Residence | | |
| | Seoul | 327 (30.2) |
| | Busan/Daegu/Ulsan/Gyeongsang | 209 (19.3) |
| | Gwangju/Jeolla/Jeju | 112 (103) |
| | Incheon/Gyeonggi/Gangwon | 329 (30.4) |
| | Daejeon/Chungcheong/Sejong | 107 (9.9) |
| Clinical experience (yearrs) | | |
| | ≤ 5 | 341 (31.5) |
| | 5–9 | 301 (27.8) |
| | 10–14 | 173 (16.0) |
| | 15–19 | 129 (11.9) |
| | 20–29 | 113 (10.4) |
| | ≥ 30 | 27 (2.5) |
| Education | | |
| | Bachelor | 677 (62.5) |
| | Master | 208 (19.2) |
| | Doctor | 199 (18.4) |
| Place of work | | |
| | Obesity treatment specialized KM clinic | 51 (4.7) |
| | KM clinic | 681 (62.8) |
| | KM hospital | 199 (18.4) |
| | Public hospital | 15 (1.4) |
| | Public community health center | 68 (6.3) |
| | Convalescent/Geriatric hospital | 46 (4.2) |
| | others | 24 (2.2) |
| Specialty | | |
| | Yes | 252 (23.2) |
| | No | 832 (76.8) |
| Specialty area of KM | | |
| | Internal medicine | 73 (29.0) |
| | Gynecology | 20 (7.9) |
| | Pediatric | 11 (4.4) |
| | Neuropsychiatry | 14 (5.6) |
| | Otolaryngology and dermatology | 15 (6.0) |
| | Rehabilitation | 52 (20.6) |
| | Acupuncture and meridian | 59 (23.4) |
| | Sasang constitutional medicine | 8 (3.2) |

All data are express in N (%). KM: Korean medicine.

**Table 2. Current status of KM clinical practice in obesity treatment.**

| | | Total (n = 1084) | Specialized in obesity treatment | | |
| --- | --- | --- | --- | --- | --- |
| | | | Specialized (n = 51) | Non-specialized (n = 1033) | P- value |
| Clinical experience in obesity treatment (years)[1] | | 6.70 (5.99) | 5.02 (4.46) | 6.78 (6.04) | **0.0089** |
| Number of obese patients treated per year[1] | | 107.13 (479.49) | 1129.47 (1679.66) | 56.65 (224.45) | **< .0001** |
| Sex of obese patients (Female, %) | | 1034 (95.4) | 51 (100) | 983 (95.2) | 0.1652 |
| Age(years) of obese patients | | | | | |
| | ≤ 19 | 18 (1.7) | 0 | 18 (1.7) | 0.1081 |
| | 20–29 | 133 (12.3) | 4 (7.8) | 129 (12.5) | |
| | 30–39 | 401 (37.3) | 27 (52.9) | 374 (36.2) | |
| | 40–49 | 387 (35.7) | 18 (35.3) | 369 (35.7) | |
| | 50–59 | 140 (12.9) | 2 (3.9) | 138 (13.4) | |
| | ≥ 60 | 5 (0.5) | 0 | 5 (0.5) | |
| Obesity level of patients* | | | | | |
| | Underweight (BMI < 18.5) | 0 | 0 | 0 | 0.6990 |
| | Normal weight (18.5 ≤ BMI ≤22.9) | 16 (1.5) | 0 | 16 (1.5) | |
| | Overweight (23 ≤ BMI ≤ 24.9) | 284 (26.2) | 13 (25.5) | 271 (26.2) | |
| | Obesity Class I (25 ≤ BMI ≤ 29.9) | 469 (43.3) | 21 (41.2) | 448 (43.4) | |
| | Obesity Class II (30 -≤ BMI ≤ 34.9) | 295 (27.2) | 15 (29.4) | 280 (27.1) | |
| | Obesity Class III (BMI > 35) | 20 (1.8) | 2 (3.9) | 18 (1.7) | |
| Comorbidities of patients[†] | | | | | |
| | Metabolic syndrome | 686 (63.3) | 30 (58.8) | 656 (63.5) | |
| | Hypertension | 420 (38.8) | 28 (54.9) | 392 (37.9) | |
| | Digestive diseases | 364 (33.6) | 25 (49.0) | 339 (32.8) | |
| | Diabetes | 327 (30.2 | 18 (35.3) | 309 (29.9) | |
| | Dyslipidemia | 308 (28.4) | 12 (23.5) | 296 (28.7) | |
| | Arthritis | 281 (25.9) | 14 (27.5) | 267 (25.8) | |
| | Cardiovascular diseases | 148 (13.7) | 8 (15.7) | 140 (13.6) | |
| | No complications | 72 (6.6) | 1 (2.0) | 71 (6.9) | |
| | Others | 51 (4.7) | 4 (7.8) | 47 (4.5) | |
| Average duration of treatment (weeks)[1] | | 4.16 (0.69) | 4.53(0.67) | 4.14(0.69) | **< .0001** |
| Visiting frequency of patient (per month)[1] | | 4.25 (3.07) | 3.45(2.85) | 4.29(3.08) | 0.0556 |
| Average length of treatment time per visit | | | | | |
| | < 10 min | 157 (14.5) | 11 (21.6) | 146 (14.1) | 0.4466 |
| | 10 ~ 20 min | 467 (43.1) | 22 (43.1) | 445 (43.1) | |
| | 20 ~ 30 min | 232 (21.4) | 10 (19.6) | 222 (21.5) | |
| | > 30 min | 228 (21.0) | 8 (15.7) | 220 (21.3) | |
| Average loss of weight by treatment[1] | | | | | |
| | ___ % of body weight | 10.56 (5.75) | 12.22 (4.12) | 10.48 (5.81) | **0.0365** |
| | Average ___ kg | 7.15 (5.60) | 7.86 (2.79) | 7.12 (5.70) | 0.3595 |

KM: Korean medicine. Data are express in N (%).

[1]: Mean (SD) [†]Multiple responses allowed. Significant P values (< 0.05) are in bold.

*The classification of obesity level by BMI is represented in the 2018 Korean Society for Study of Obesity Guideline for the management of obesity in Korea. This classification is in accordance with the WHO obesity guideline for the Asia-Pacific region.

(n = 473) reported using the KM Syndrome Differentiation Questionnaire for Obesity [21]. As the primary outcome indicators, BMI, body weight, and body fat percentage were used by approximately 60% of total respondents; however, respondents in obesity-specialized clinics reported using body fat percentage as the most common indicator (n = 40, 78.4%). Almost half

**Table 3. Diagnostic tools and therapeutic intervention in KM for obesity treatment.**

| | | Total (n = 1084) | Specialized in obesity treatment | |
| --- | --- | --- | --- | --- |
| | | | Specialized (n = 51) | Non-specialized (n = 1033) |
| Diagnostic tool or device[†] | | | | |
| | Bioelectric impedance (e.g. InBody[TM]) | 925 (85.3) | 50 (98.0) | 875 (84.7) |
| | KM Syndrome Differentiation Questionnaire for Obesity | 473 (43.6) | 23 (45.1) | 450 (43.6) |
| | Body thermometer | 97 (9.0) | 6 (11.8) | 91 (8.8) |
| | Ryodoraku analyzer | 52 (4.8) | 1 (2.0) | 51 (4.9) |
| | Pulse diagnosis instrument | 43 (4.0) | 3 (5.9) | 40 (3.9) |
| | Tongue diagnosis instrument | 18 (1.7) | 1 (2.0) | 17 (1.6) |
| | Others | 21 (1.9) | 2 (3.9) | 19 (1.8) |
| | None | 80 (7.4) | 0 | 80 (7.7) |
| Primary outcome indicator[†] | | | | |
| | Body mass index (BMI) | 654 (60.3) | 23 (45.1) | 631 (61.1) |
| | Body weight | 636 (58.7) | 34 (66.7) | 602 (58.3) |
| | Percentage of body fat | 629 (58.0) | 40 (78.4) | 589 (57.0) |
| | Abdominal fat rate | 227 (20.9) | 11 (21.6) | 216 (20.9) |
| | Waist circumstance | 180 (16.6) | 3 (5.9) | 177 (17.1) |
| Use of KM syndrome differentiation for diagnosis (Yes, %)[*] | | 558 (51.5) | 16 (31.4) | 542 (52.5) |
| Diagnostic type (n = 558) | Eight principle pattern identification | 166 (29.7) | 5 (31.3) | 162 (29.8) |
| | Organ system diagnosis | 182 (32.6) | 4 (25.0) | 179 (33.0) |
| | Defensive qi and nutrient blood diagnosis | 10 (1.8) | 1 (6.3) | 9 (1.7) |
| | Sasang constitutional medicine diagnosis | 182 (32.6) | 5 (31.3) | 177 (32.6) |
| | Meridian system diagnosis | 13 (2.3) | 0 | 13 (2.4) |
| | Six meridian diagnosis | 23 (4.1) | 0 | 23 (4.2) |
| | Diagnostic type by KM Obesity CPG[††] | 218 (39.1) | 9 (56.3) | 209 (38.5) |
| | Others | 6 (1.1) | 0 | (1.1) |
| Primary factor for deciding therapeutic intervention[†] | | | | |
| | Lifestyle habits (e.g. exercise, eating habits and nutritional status) | 790 (72.9) | 39 (76.5) | 751 (72.7) |
| | Obesity level (e.g. body weight, BMI) | 777 (71.7) | 43 (84.3) | 734 (71.1) |
| | Purposes of treatment (e.g. weight loss, body shape) | 434 (40.0) | 18 (35.3) | 416 (40.3) |
| | Medical history or complications | 276 (25.5) | 12 (23.5) | 264 (25.6) |
| | Age | 271 (25.0) | 13 (25.5) | 258 (25.0) |
| | Duration of treatment | 210 (19.4) | 13 (25.5) | 197 (19.1) |
| | Sasang constitution | 199 (18.4) | 6 (11.8) | 193 (18.7) |
| | Economic factor | 109 (10.1) | 5 (9.8) | 104 (10.1) |
| Treatment methods for obesity[†] | | | | |
| | Herbal medicine | 1037 (95.7) | 49 (96.1) | 998 (95.6) |
| | Electroacupuncture | 532 (49.1) | 22 (43.1) | 510 (49.4) |
| | Control diet (e.g. fasting, caloric restriction) | 506 (46.7) | 21 (41.2) | 485 (47.0) |
| | Lifestyle intervention for obesity | 494 (45.6) | 27 (52.9) | 467 (45.2) |
| | Acupuncture | 399 (36.8) | 14 (27.5) | 385 (37.3) |
| | Pharmacoacupuncture | 214 (19.7) | 14 (27.5) | 200 (19.4) |
| | Cupping | 150 (13.8) | 5 (9.8) | 145 (14.0) |
| | Moxibustion | 110 (10.2) | 4 (7.8) | 106 (10.3) |
| | Chuna | 46 (4.2) | 3 (5.9) | 43 (4.2) |
| | Qigong | 2 (0.2) | 1 (2.0) | 1 (0.1) |

(*Continued*)

**Table 3.** (Continued)

| | | Total (n = 1084) | Specialized in obesity treatment | |
| | | | Specialized (n = 51) | Non-specialized (n = 1033) |
|---|---|---|---|---|
| | Others | 16 (1.5) | 2 (3.9) | 14 (1.4) |

KM: Korean medicine. All data are express in N (%).

[†]Multiple responses allowed.

[*]P-value = 0.0033 (between groups).

[††]Spleen deficiency pattern, Food accumulation pattern, Phlegm-fluid retention pattern, Liver depression pattern, Yang deficiency pattern, Static blood pattern.

of the total respondents were using KM syndrome differentiation for diagnosis, but its usage was significantly low in specialized clinics (n = 16, 31.4%) ($p < 0.01$). Among the diagnostic type of KM syndrome differentiation, the diagnostic type according to the KM Obesity Clinical Practice Guideline was the most used (n = 218, 39.1%), and the same result was noted in the subgroup analysis. A similar number of KMDs responded that lifestyle habits (n = 790, 72.9%) or obesity level (n = 777, 71.7%) were the primary factors used for determining interventions for obesity. Almost all respondents were using herbal medicine for treating obesity and patient satisfaction was also the highest with herbal medicine (S1 Table). In addition, more respondents were using electroacupuncture (n = 532, 49.1%) than normal acupuncture (n = 399, 36.8%) for obesity treatment, and almost half of the respondents were teaching lifestyle interventions (n = 506, 46.7%) or diet control (n = 506, 46.7%).

### Usage pattern of herbal medicine for obesity treatment

As shown in Table 3, almost all responded KMDs were prescribing herbal medicine for obesity treatment. Some clinical trials have reported that herbal medicines are effective in weight loss and improvement of blood lipid profiles [22–24]. However, it has been not revealed in real clinical field about herbal medicines or herbs prescribed for obesity and the amount of prescription. Table 4 demonstrates frequently prescribed herbal medicines in obesity treatment. Taeeumjowui-tang is the most prescribed herbal formula (n = 546, 50.4%), also noted in the sub-group analysis. Of all the respondents, 76.7% (n = 831) reported that they add a specific herb for obesity regardless of the herbal formula or syndrome differentiational diagnosis, and more than 80% of the respondents reported using Ephedrae Herba the most (Table 4).

The responded KMDs reported that they prescribed herbal medicine for an average 8.5 weeks to patients. KMDs in obesity-specialized clinics reported prescribing herbal medicine (10.22 ± 4.58 weeks) for significantly longer duration than those in non-specialized clinics/hospitals (8.42±3.98 weeks) ($p < 0.01$). Almost all KMDs (n = 1042, 96.1%) reported the use of Ephedrae Herba for obesity treatment, and its prescription dose was 5.18 ± 2.7 g/day at minimum and 10.06 ± 4.23 g/day at maximum. Moreover, respondents from obesity-specialized clinics reported prescribing significantly more amount of Ephedrae Herba than KMDs in non-specialized clinics/hospitals at both minimum and maximum doses ($p < 0.0001$). As the primary determinant for its dose, caffeine sensitivity was considered by most respondents (n = 817, 78.4%), followed by obesity level (n = 630, 60.5%), and sleeping habits (n = 415, 39.8%).

In addition, 62.9% of responded KMDs answered that at least one of the patients with who were prescribed Ephedrae Herba had ever experienced discomfort symptoms, in the order of neuropsychiatric, gastrointestinal and cardiovascular events. In contrast, only 23.2% of respondents reported the occurrence of discomfort symptoms by herbal medicines not

**Table 4. Usage pattern of herbal medicine for obesity treatment.**

| | | Total (n = 1084) | Specialized in obesity treatment | | |
| --- | --- | --- | --- | --- | --- |
| | | | Specialized (n = 51) | Non-specialized (n = 1033) | P- value |
| Average administration duration of herbal medicine (week)[1] | | 8.50 (4.02) | 10.22 (4.58) | 8.42 (3.98) | **0.0018** |
| Frequently prescribed herbal formula[†] | | | | | |
| | Taeeumjowui-tang | 546 (50.4) | 26 (51.0) | 520 (50.3) | |
| | Gambihwan | 378 (34.9) | 19 (37.3) | 359 (34.8) | |
| | Euiiyin-tang | 348 (32.1) | 9 (17.6) | 339 (32.8) | |
| | Bangpungtongseong-san (Bofutsushosan) | 258 (23.8) | 6 (11.8) | 252 (24.4) | |
| | Jowiseungcheung-tang | 172 (15.9) | 3 (5.9) | 169 (16.4) | |
| | Bangkihwangki-tang (Boiogito) | 92 (8.5) | 2 (3.9) | 90 (8.7) | |
| | Gamrosu | 55 (5.1) | 1 (2.0) | 54 (5.2) | |
| | Buhnsimgieum | 42 (3.9) | 1 (2.0) | 41 (4.0) | |
| | Cheongpyesagan-tang | 40 (3.7) | 0 | 40 (3.9) | |
| | Anmyungambi-tang | 9 (0.8) | 0 | 9 (0.9) | |
| | Others | 154 (14.2) | 9 (17.6) | 145 (14.0) | |
| Frequently used herbs (regardless of herbal formula or syndrome differentiation) (Yes, %) | | 831 (76.7) | 40 (78.4) | 791 (76.6) | 0.7594 |
| Top 10 ranked[†#] | Ephedrae Herba | 658 (81.0) | 33 (84.6) | 625 (80.9) | |
| | Coicis Semen | 376 (46.3) | 11 (28.2) | 365 (47.2) | |
| | Gypsum Fibrosum | 88 (10.8) | 2 (5.1) | 86 (11.1) | |
| | Rehmanniae Radix Preparata | 39 (4.8) | 1 (2.6) | 38 (4.9) | |
| | Poria Sclerotium | 37 (4.6) | 2 (5.1) | 35 (4.5) | |
| | Rhei Radix et Rhizoma | 28 (3.4) | 1 (2.6) | 27 (3.5) | |
| | Alismatis Rhizoma | 22 (2.7) | 0 | 22 (2.8) | |
| | Astragali Radix | 17 (2.1) | 0 | 17 (2.2) | |
| | Atractylodis Rhizoma | 16 (2.0) | 0 | 16 (2.1) | |
| | Angelicae Gigantis Radix | 15 (1.8) | 0 | 15 (1.9) | |
| Patient complaining discomfort symptoms by herbal medicine not containing Ephedrae Herba | | | | | |
| | Yes, % | 251 (23.2) | 16 (31.4) | 235 (22.7) | 0.1541 |
| | No, % | 833 (76.8) | 35 (68.6) | 798 (77.3) | |
| Uncomfortable symptoms by herbal medicine not containing Ephedrae Herba[†] (n = 251) | | | | | |
| | Neuropsychiatric[a] | 107 (9.9) | 5 (9.8) | 102 (9.9) | |
| | Gastrointestinal[b] | 120 (11.1) | 6 (11.8) | 114 (11.0) | |
| | Cardiovascular[c] | 57 (5.3) | 2 (3.9) | 55 (5.3) | |
| | Abnormal level on blood test[d] | 34 (3.1) | 5 (9.8) | 29 (2.8) | |
| | Dermatological[e] | 42 (3.9) | 4 (7.8) | 38 (3.7) | |
| | Geniourinary | 24 (2.2) | 2 (3.9) | 22 (2.1) | |
| | Musculoskeletal | 7 (0.6) | 0 | 7 (0.7) | |
| | Respiratory | 1 (0.1) | 0 | 1 (0.1) | |
| | Severe adverse events[f] | 0 | 0 | 0 | |
| | Others | 24 (2.2) | 4 (7.8) | 20 (1.9) | |
| Do you prescribe Ephedrae Herba for obesity treatment? (Yes, %) | | 1042 (96.1) | 51 (100) | 991 (95.9) | 0.2572 |
| Prescription dose of Ephedrae Herba[1] (n = 1042) | | | | | |
| | Minimum (g/day) | 5.18 (2.70) | 6.59 (3.04) | 5.11 (2.67) | < **.0001** |
| | Maximum (g/day) | 10.06 (4.23) | 12.67 (3.91) | 9.92 (4.20) | < **.0001** |
| Primary factor for deciding dose of Ephedrae Herba[†] (n[g] = 1042) | | | | | |
| | Caffeine sensitivity (e.g. Heart palpitation) | 817 (78.4) | 41 (80.4) | 776 (78.3) | |

*(Continued)*

**Table 4.** (Continued)

| | | Total (n = 1084) | Specialized in obesity treatment | | |
| --- | --- | --- | --- | --- | --- |
| | | | Specialized (n = 51) | Non-specialized (n = 1033) | P- value |
| | Obesity level | 630 (60.5) | 35 (68.6) | 595 (60.0) | |
| | Sleeping habits | 415 (39.8) | 18 (35.3) | 397 (40.1) | |
| | Sasang constitutional type | 272 (26.1) | 11 (21.6) | 261 (26.3) | |
| | Others | 48 (4.6) | 5 (9.8) | 43 (4.3) | |
| Patient complained uncomfortable symptoms by herbal medicine containing Ephedrae Herba ($n^g$ = 1042) | | | | | |
| | Yes, % | 655 (62.9) | 36 (70.6) | 619 (62.5) | 0.2415 |
| | No, % | 387 (37.1) | 15 (29.4) | 372 (37.5) | |
| Notice of caution for caffeine consumption (Yes, %) ($n^h$ = 655) | | 605 (92.4) | 35 (97.2) | 570 (92.1) | 0.5115 |
| Uncomfortable symptoms by herbal medicine containing Ephedrae Herba[†] | | | | | |
| | Neuropsychiatric[a] | 419 (40.2) | 20 (39.2) | 399 (40.3) | |
| | Gastrointestinal[b] | 400 (38.4) | 26 (51.0) | 374 (37.7) | |
| | Cardiovascular[c] | 240 (23.0) | 11 (21.6) | 229 (23.1) | |
| | Abnormal level on blood test[d] | 36 (3.5) | 4 (7.8) | 32 (3.2) | |
| | Dermatological[e] | 46 (4.4) | 8 (15.7) | 38 (3.8) | |
| | Geniourinary | 51 (4.9) | 2 (3.9) | 49 (4.9) | |
| | Musculoskeletal | 4 (0.4) | 0 | 4 (0.4) | |
| | Respiratory | 2 (0.2) | 0 | 2 (0.2) | |
| | Severe adverse events[f] | 0 | 0 | 0 | |
| | Others | 31 (3.0) | 1 (2.0) | | |

Data are express in N (%).

[1]: Mean (SD).

[†]Multiple responses allowed.

[#]n = 831.

[a] e.g. anxiety, insomnia, depression, vision decrease.

[b] e.g. nausea, dry mouth, vomiting.

[c] e.g. tachycardia, palpitation.

[d] e.g. ALT, AST, Creatinine, BUN.

[e] e.g. rash, urticarial

[f] e.g. death, myocardiac infarction, stroke, seizure.

[g] Specialized (n = 51), Non-specialized (n = 991)

[h] Specialized (n = 36), Non-specialized (n = 619). Significant P values (< 0.05) are in bold.

containing Ephedrae Herba, in the order of gastrointestinal (n = 120, 11.1%), neuropsychiatric (n = 107, 9.9%) and cardiovascular events (n = 57, 5.3%).

## Post-treatment management for obesity

Despite obesity treatment or efforts to achieve weight loss, only a few obese people could maintain reduced weight for a long-term. Thus, a sub-group analysis was conducted according to the responses to the status of post-treatment management (Table 5). Post-treatment management after taking herbal medicines for weight loss may include consultation via telephone and additional supportive care, such as acupuncture, psychological support, teaching for diet control and physical activity. Even after herbal medicine treatment for weight loss, 677 of 1084 respondents were providing management care, and 47.6% (n = 322) of the 677 respondents

**Table 5. Post-treatment management for obesity treatment.**

| | | Total (n = 1084) | Specialized in obesity treatment | | |
|---|---|---|---|---|---|
| | | | Specialized (n = 51) | Non-specialized (n = 1033) | P- value |
| Post-treatment managements (including phone call consulting) (Yes, %) | | 677 (62.4) | 42 (82.3) | 635 (61.5) | |
| Post-treatment managements duration (n = 677) | | | | | |
| | < 2 weeks | 79 (11.7) | 6 (14.3) | 73 (11.5) | < .0001 |
| | 2 weeks—1 months | 150 (22.2) | 6 (14.3) | 144 (22.7) | |
| | 1 ~ 3 months | 322 (47.6) | 14 (33.3) | 308 (48.5) | |
| | 3 ~ 6 months | 79 (11.7) | 4 (9.5) | 75 (11.8) | |
| | ≥ 6 months | 47 (6.9)) | 12 (28.6) | 35 (5.5) | |
| | | Total (n = 1084) | Post-treatment managements | | |
| | | | Yes (n = 677) | No (n = 407) | P- value |
| Average weight loss[1] | ___ % of body weight | 10.56 (5.75) | 11.23 (6.61) | 9.44 (3.67) | < .0001 |
| | Average ___ kg | 7.15 (5.60) | 7.54 (5.84) | 6.50 (5.10) | **0.0030** |
| Do you prescribe Ephedrae Herba for obesity treatment? (Yes, %) | | 1042 (96.1) | 653 (96.5) | 389 (95.6) | 0.4685 |
| Prescription dose of Ephedrae Herba[1] (n = 1042) | | | | | |
| | Minimum (g/day) | 5.18 (2.70) | 5.08 (2.77) | 5.35 (2.57) | 0.1140 |
| | Maximum (g/day) | 10.06 (4.23) | 9.87 (4.34) | 10.37 (4.02) | 0.0684 |

Data are express in N (%).

[1]: Mean (SD). Significant P values (< 0.05) are in bold.

reported the post-treatment management duration to be from 1 to 3 months, while 22.2% (n = 150) reported 2 weeks to 1 month. In obesity-specialized KM clinics, 82.3% of respondents (n = 42) reported providing management care after treatment. A larger proportion of KMDs in specialized clinics than those in non-specialized clinics/hospitals were providing long-term management care for more than 6 months. Table 5 demonstrates that the average weight loss by treatment in both indexes was significantly more weight loss in the KMDs group providing post-treatment management than group that did not provide this management. However, there are no significant differences in prescription dose of Ephedrae Herba.

## Discussion

KM is characterized by diagnosis and treatment based on traditional methods, such as syndrome differentiation diagnosis, herbal medicine, acupuncture and moxibustion. KM is recognized as the double axis of health care system of South Korea along with conventional western medicine. Thanks to the National Health Insurance, patients can easily receive suitable KM treatment according to their illness, such as musculoskeletal and gastrointestinal disease; however, treatments for obesity are rarely covered. For this reason, basic information on the grades of obesity usually treated with KM has not been collected as official medical insurance statistics. The present study revealed that 43.3% of KMD respondents (n = 469) reported that the average obesity level of the patients treated was Obesity Class I (25 ≤ BMI ≤ 29.9), followed by Obesity Class II (BMI 30 ~ 34.9, 27.2%, n = 295) and overweight patients (23 ≤ BMI ≤ 24.9, 26.2%, n = 284). Furthermore, Cheon and Jang reported that the proportion of patients taking herbal medicine according to BMI grade based on survey data [11]. They indicated that among those taking herbal medicine as a weight control strategy, 51.7% had a BMI of less than 25, 33.2% had a BMI of 25–29.9, and 15.1% had a BMI of over 30. Given the result of the present study that almost all KMDs prescribed herbal medicines for weight loss, it is considered

that KM is usually conducted with overweight and Obesity Class I patients. Nevertheless, several studies reported that herbal medicines have a more positive weight loss effect for the patients with higher initial BMI [25, 26]. Unlike herbal medicine, acupuncture may be more effective for overweight patients than for obese patients [10]. From this point of view, acupuncture combined with lifestyle modification could provide a promising therapeutic option to those not requiring pharmacotherapy and bariatric surgery. However, obesity level for which KM treatment is most effective needs to be more elucidated. Even normal weight population are received KM treatment for aesthetic purposes, and although some studies have been conducted on the effect of KM treatment on severe obesity (BMI over 30 kg/m$^2$) [26, 27], additional research is needed on the effects of KM treatment on population groups of specific obesity levels.

While acupuncture is the most commonly used treatment for patients generally [28], the findings of the present study indicate that the most commonly used treatment method for obesity is herbal medicine. In addition, the patients, who have ever treated in KM clinic/hospital, reported that herbal medicine is not easily accessible among KM treatments in terms of its costs and most needed the coverage of the National Health Insurance [28]. The present study revealed that most of the responded KMDs reported complaints about the financial burden on patients for medical costs as the major difficulty in obesity treatment (S2 Table). Because the real clinical fields of KM treatments for obesity has not been well documented, real world information or statistic data are poor. The present study is the first to elucidate current clinical practice pattern of KM for obesity treatment by surveyed investigation.

The obesity treatment process requires continuous and comprehensive care, including education, diet control, exercise therapy, and drug medication. The most important therapeutic strategy for obesity is weight control, and it has also been elucidated that more weight loss induces more clinical improvement. Several studies have also reported that weight loss, even moderate (5–10% of body weight), is associated with improvement in related symptoms and comorbidities, as well as reduction in medical costs [29–31] However, the KNHNES 2015 indicated that 33% of all subjects attempted to lose weight for 1 year, but only 15.4% successfully achieved weight loss [32]. Likewise, the success rate of losing weight is low compared to the efforts for weight loss, thereby warranting appropriate intervention strategies for weight loss. The present study found that KMDs were providing medical care for the average patient model woman in the 30s with Obesity Class I ($25 \leq$ BMI $\leq 29.9$). The patient could lose 10.56% of body weight (7.15 kg) by taking herbal medicine mainly containing Ephedrae Herba (5.18–10.06 g/day) for 8.5 weeks with electroacupuncture. Moreover, the responded KMDs reported that the highest satisfaction was with herbal medicine treatment (S1 Table). A systematic review also reported that taking herbal medicine and acupuncture are more effective than placebo or lifestyle modification in body weight reduction [33]. The weight reduction effects of herbal formulas represented in this survey were also well established in several randomized controlled trials. Taeeumjowui-tang showed significant clinical improvements in body weight, waist-circumference, waist hip ratio, total cholesterol and LDL-cholesterol of obese patients by 12 weeks medication, compared to placebo [16]. Bangpungtongseong-san (Bofutsusho-san) significantly reduced body weight and BMI of obese patients (BMI > 25 kg/m$^2$) [14], decreased visceral fat and improved insulin resistance in obese women with impaired glucose tolerance [22], and its anti-obesity properties were associated with polymorphisms in obesity-related genes [13]. Euiiyin-tang has potential weight loss effects in obese women, but does not affect lipid profiles [15]. Furthermore, in obesity treatment along with weight control, it is necessary to improve the metabolic indicators, such as blood pressure, blood glucose and lipid profiles. Although some cellular, animal studies and clinical trials have reported improvement in lipid profiles by herbal medicine [24, 34], this aspect has not been fully elucidated. Because

KMD is difficult to access blood test for diagnostic purpose in the real clinical field, the effects of herbal medicine on lipid profiles are difficult to evaluate in the real world of KM clinical practice, as was also difficult in the present study.

In this survey, almost all KMDs were found to prescribe Ephedrae Herba for obesity treatment, and almost all herbal formulas except Bangkihwangki-tang and Buhnsimgieum, contained Ephedrae Herba (Table 4). Ephedrae Herba is defined as the dried terrestrial stem of *Ephedra sinica* Stapf or other ephedrine-containing *Ephedra* species and alkaloid. Its weight reduction effect is recognized to result from suppressing appetite and promoting the metabolic rate of adipose tissue [17]. FDA allows 150 mg/day of ephedrine for medical use and The Society of Korean Medicine for Obesity Research recommends using dried Ephedrae Herba 4.5–7.5 g/day for up to 6 months [35]. Mills et al suggested dried Ephedra Herba 3–12 g/day as regular dose and ephedrine 60–90 mg/day for obesity treatment [36]. The present study discovered that KMDs prescribe Ephedrae Herba 5.18 g/day at minimum and 10.06 g/day at maximum dose for obesity treatment, which is at least 36.12–70.42 mg of the total alkaloid as ephedrine or pseudoephedrine that estimated to 0.7% by Korean Pharmacopoeia (12nd Ed.). Although it is regarded to be within the safety range, the effects of its cumulative use have still not have established. Therefore, further studies regarding chemical analysis and safety test using actual prescription herbal medicine are needed for the safety long-term use of Ephedrae Herba.

Although the anti-obesity properties of Ephedrae Herba (*Ma huang*) and ephedrine are well elucidated, their health risks also exist, especially when consumed in combination with caffeine [37]. In the present study, there were three times more KMDs who ever experienced patients complaining uncomfortable symptoms by herbal medicine containing Ephedrae Herba than by herbal medicine not containing it. Meanwhile almost all the KMDs with these experiences considered the relation with caffeine and also notified about them to their patients when prescribing herbal medicine containing Ephedrae Heba, which indicates that KMDs are well aware of the possible adverse drug reactions caused by Ephedrae Herba. Ephedrae Herba can reduce fatigue and lessened desire for sleep in a short-term use but can also cause anxiety, restlessness and insomnia when used at higher dose [38]. Interestingly, KMDs reported more neuropsychiatric or gastrointestinal events caused by herbal medicine containing Ephedrae Herba than cardiovascular events which were the most widely known drug reaction by Ephedrae Herba [19]. Moreover, approximately 40% of the responded KMDs reported sleeping habits as a determinant for the dose of Ephedra Herba. These results indicate that KMDs recognize and consider the psychiatric effects of Ephedrae Herba. Some reports on adverse event related ephedra and ephedrine had been submitted to the U.S. FDA, such as hypertension, palpitations, tachycardia and stroke; however, it may be attributed to misuse, abuse, contraindication, hypersensitivity or drug interaction [17]. According to the review on case report files with the FDA, the majority of case reports are not sufficient to prove a causal relationship between the use of ephedra and ephedrine and the adverse event in question [39]. Furthermore, adverse events complained by the participants in clinical trials were only mild sympathetic excitation symptoms such as insomnia, anxiety, nervous sensitivity, shaking hands, palpitations, nausea, vomiting, constipation, dry mouth, headache, and dizziness, but no significant changes on blood/urine test, no cardiovascular side effects, and no severe adverse events were noted [34]. Although there exists a limitation that the clinical trials with herbal medicine did not enroll sufficient number of patients for detecting serious adverse drug reactions, KMDs rarely prescribed Ephedrae Herba alone, but prescribed it as herbal formulas, in combination with other medicinal herbs, which might contribute to reduce adverse drug reactions and allow safe and long-term use of herbal medicine.

In the present study, subgroup analysis was conducted by dividing the working place of respondents into obesity-specialized clinics and non-specialized clinics/hospitals. Obesity-specialized KM clinics are characterized by provision medical care focusing on obesity treatment rather than on musculoskeletal treatment which accounts for the majority of other KM clinics/hospitals. The result of this study showed that KMDs working in obesity-specialized clinics, compared with those working in non-specialized clinics/hospitals, reported significant larger number of obese patients, longer treatment duration, longer post-treatment management duration, longer average administration duration of herbal medicine, more prescription dose of Ephedrae Herba, and more average weight loss of patients, but significantly lesser use of KM syndrome differentiation for diagnosis.

Bariatric surgery has been covered by the National Health Insurance since 2019 in Korea, and it is very effective for losing weight and the improving obesity-related comorbidities [40, 41]. In addition, lower body lift, which restore body contour in postbariatric patients, could improve overall quality of life by resulting in pleasant aesthetic outcomes for the obese patients [42]. Despite these advantages, bariatric surgery is only employed for severely obese patients, and postoperative surgical complications due to invasive surgical methods still remains. Moreover, decreased stomach capacity and oral intake, and hormonal changes due to the surgery could cause malabsorption that leads to several nutritional deficiencies requiring long-term supplementation [43]. In particular, in post-menopausal women with obesity, these nutrient deficiencies could increase the risk of skeletal muscle weakness, and metabolic bone disease leading to bone loss and predisposition to fracture [43–45]. Even if weight loss is achieved, its long-term results are generally poor. As a physiological response to weight loss, our body changes various hormones and energy consumption. Hence, control food intake and energy expenditure for maintaining weight loss and preventing weight regain is still challenged after weight reduction. Diet control, regular physical activities and frequent weight monitoring are necessary for long-term weight maintenance [46]. However, sustaining reduced weight can be difficult with lifestyle changes alone, and additional medical therapies may help maintain weight loss [46]. In contrast to bariatric surgery or conventional drugs for obesity, KM, which is represented by treatment using herbal medicine and acupuncture, has the advantage that it can be practiced in parallel with lifestyle modification, and it is associated fewer adverse events. Acupuncture, including not only manual acupuncture but also auricular acupuncture, electroacupuncture, pharmacopuncture, and catgut embedding, effectively treats overweight/obesity by suppressing appetite and relieving hunger and fatigue during weight loss, and its effects on weight loss may be maximized when it combined with lifestyle modification [9, 10]. Subgroup analysis in the present study revealed that despite no significant differences noted in the prescription dose of Ephedrae Herba, the average weight loss of obese patients was significantly greater in the group of KMDs who provided maintenance care after prescribing herbal medicine. These results suggest that obese patients can successfully accomplish weight loss in the long-term through post-treatment management care. Herbal medicine including medicinal herbs, their active compounds, and mixed herbal preparations, has beneficial effects on obesity, such as weight loss, waist-hip ratio reduction, body fat reduction, and food intake reduction [8, 23], and is effective for obesity compared with conventional drugs, placebos or lifestyle control [7]. In addition, it has been reported its pharmacological properties, such as controlling appetite, inhibiting pancreatic lipase activity, stimulating thermogenesis and lipid metabolism, increasing satiety, promoting lipolysis, regulating adipogenesis, and inducing apoptosis in adipocytes [47]. Given these anti-obesity properties of acupuncture and herbal medicine, KM treatment may help to alleviate these adaptive physiological responses after weight loss. KM treatment generally focuses on clinical symptoms and subjective signs, such as overall body conditions, digestive symptoms, pain, and sleeping habit. Thus, herbal medicine and

acupuncture may reduce side effects of pharmacotherapy and bariatric surgery. Additional studies are also required to evaluate the efficacy and safety of herbal medicine and acupuncture in combination with other approved conventional drugs for obesity and bariatric surgery.

However, there is currently insufficient evidence to recommend several herbal medicines for weight loss due to the low quality or small sample size of the clinical trial [7, 48]. Although the safety and effectiveness of KM treatment for obesity is recognized, further large-scale and long-term clinical trials need to be conducted for evaluating KM treatment. In addition, the active ingredients in herbs as well as its molecular targets and underlying mechanisms of action should also be determined. In particular, a direct causal relationship between Ephedae Herba, the herb most frequently prescribed for weight loss by KMDs in this study, and reported adverse events has not been established, thus further safety studies should clarify potential adverse reactions. KM is based on prescribing herbal medicines according to KM syndrome differentiation diagnosis, but only half of the KMDs in the present study used KM syndrome differentiation diagnosis for obesity. Han *et al*. warned that the administration of herbal medicines not based on KM syndrome differentiation diagnosis for obesity may contribute to the occurrence of adverse reactions based on the results of study [8]; thus, more attention should be paid to the use of herbal medicine for weight loss.

Taken together, this study represents the current clinical practice pattern of KM for obesity by using a cross-sectional survey, including the diagnosis, prognosis, and treatment. Especially, the current prescription dose of Ephedrae Herba, which is the most frequently prescribed herb for obesity treatment, by KMDs for obesity treatment, was first determined in this study, and the dose was found to be greater in obesity-specialized KM clinics than in non-specialized clinics/hospitals. One of the limitations of this study is that it did not elicit the success rate of weight loss and the incidence of side effects by KM treatment for obesity. Because this study was based on a self-report survey, there is a possibility of bias, such as insincere responses, exaggerated treatment effectiveness, and distortion in the number of patients and in the occurrence of adverse events. Nevertheless, it has significance in that it is the first national survey showing the current clinical practice pattern of KMDs for treating obesity. The overall effectiveness and safety of KM treatment for obesity has been well documented through various study designs, such as retrospective studies, systematic reviews, and meta-analyses [7, 8, 10, 26, 33]. According to the retrospective review of 124 patients who had taken Gamitaeeumjowee-tang for 10 weeks, the overall rate of adverse events was 37.1% during Week 2–4 and 16.9% at Week 10 (for causality of adverse events using the WHO-Uppsala Monitoring Centre causality categories, 52.2% were evaluated as "possible" at Week 2–4 and 57.1% were evaluated as "unlikely" at Week 10) [49]. It is not possible to establish the direct causal relationship between the degree of weight loss and management care after taking herbal medicine and between the prescribed amount of Ephedrae Herba and its adverse drug reaction in this study. Although the average prescription dose of Ephedrae Herba is considered safe based on previous reports and recommendations, additional studies are necessary to ensure the safe and long-term use of herbal medicine that contains Ephedrae Herba. It is essential to build a prospective registry of herbal medicine for weight loss to register herbal medicine, and any side effects, including their causality and severity. This will enable the acute statistical investigation of the occurrence of side effects. In conclusion, the findings showed the actual usage pattern of herbal medicine for obesity treatment, which was not revealed due to exclusion in health insurance. This data would be valuable in making a clinical pathway of obesity care in KM clinics and hospitals reflecting real clinical practice.

## Supporting information

**S1 Table. Patient satisfaction with treatment for obesity.**
(DOCX)

**S2 Table. Difficulties in obesity treatment.**
(DOCX)

**S1 Appendix. Questionnaire: A national survey on current clinical practice pattern of Korean Medicine doctors for treating obesity.**
(PDF)

## Author Contributions

**Conceptualization:** Kyung Hwan Jegal, Mi Mi Ko, Bo-Young Kim, Mi Ju Son, Sungha Kim.

**Data curation:** Kyung Hwan Jegal, Mi Mi Ko, Bo-Young Kim.

**Formal analysis:** Mi Mi Ko.

**Funding acquisition:** Sungha Kim.

**Investigation:** Kyung Hwan Jegal, Bo-Young Kim.

**Methodology:** Kyung Hwan Jegal, Mi Ju Son, Sungha Kim.

**Project administration:** Sungha Kim.

**Supervision:** Sungha Kim.

**Writing – original draft:** Kyung Hwan Jegal.

**Writing – review & editing:** Kyung Hwan Jegal, Mi Ju Son, Sungha Kim.

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
