## [Decision Letter · Decision Letter 0]

28 Sep 2021

PONE-D-21-26587A national survey on current clinical practice pattern of Korea medicine doctors for treating obesityPLOS ONE

Dear Dr. Kim,

Thank you for submitting your manuscript to PLOS ONE. After careful consideration, we feel that it has merit but does not fully meet PLOS ONE’s publication criteria as it currently stands. Therefore, we invite you to submit a revised version of the manuscript that addresses the points raised during the review process.

The paper needs major revisions.==============================

We look forward to receiving your revised manuscript.

Kind regards,

Alessandro de Sire, M.D.

Academic Editor

PLOS ONE

Journal Requirements:

a) Did participants provide their written or verbal informed consent to participate in this study?

3. Please include your tables as part of your main manuscript and remove the individual files. Please note that supplementary tables (should remain/ be uploaded) as separate "supporting information" files.

4. Please include additional information regarding the survey or questionnaire used in the study and ensure that you have provided sufficient details that others could replicate the analyses. For instance, if you developed a questionnaire as part of this study and it is not under a copyright more restrictive than CC-BY, please include a copy, in both the original language and English, as Supporting Information.

“SK received funding from the Ministry of Health & Welfare, Republic of Korea (Grant No.: HF20C0208)”

“This research was supported by a grant of the Korea Health Technology R&D Project through the Korea Health Industry Development Institute (KHIDI), funded by the Ministry of Health & Welfare, Republic of Korea (Grant No.: HF20C0208)”

We note that you have provided funding information within the Acknowledgements Section. Please note that funding information should not appear in the Acknowledgments section or other areas of your manuscript. We will only publish funding information present in the Funding Statement section of the online submission form.

“SK received funding from the Ministry of Health & Welfare, Republic of Korea (Grant No.: HF20C0208)”

Reviewers' comments:

Reviewer's Responses to Questions

**Comments to the Author**

1. Is the manuscript technically sound, and do the data support the conclusions?

Reviewer #1: Yes

Reviewer #2: No

2. Has the statistical analysis been performed appropriately and rigorously? 

Reviewer #1: Yes

Reviewer #2: N/A

3. Have the authors made all data underlying the findings in their manuscript fully available?

Reviewer #1: No

Reviewer #2: Yes

4. Is the manuscript presented in an intelligible fashion and written in standard English?

Reviewer #1: Yes

Reviewer #2: No

5. Review Comments to the Author

Reviewer #1: Dear Authors, the study is interesting and this is an emerging topic. However, There are some

issues that should be addressed with a reply point-by-point

Which grade of obesity is usually treated with KM? please address with appropriate bibliography.

Is the KM first line treatment or alternative treatment? for which obesity grade/s.? Please clarify

Line 262. Obesity class I is >30 kg/m2 in western literature. Please clarify your statement and table ; moreover, please provide appropriate bibliography

Bariatric Surgery is gaining popularity in western medicine for treating obesity. The key factor for such growing popularity is the improvement in obesity-related comorbidities. On the other hand bariatric surgery could cause malabsorption that leads to several nutritional deficiencies requiring long-term supplementation. Could you find such pros and cons in KM?

Please expand the discussion mentioning the listed studies.

Losco L, Roxo AC, Roxo CW, Lo Torto F, Bolletta A, de Sire A, Aksoyler D, Ribuffo D, Cigna E, Roxo CP. Lower Body Lift After Bariatric Surgery: 323 Consecutive Cases Over 10-Year Experience. Aesthetic Plast Surg. 2020 Apr;44(2):421-432. doi: 10.1007/s00266-019-01543-x.

Gimigliano F, Moretti A, de Sire A, Calafiore D, Iolascon G. The combination of vitamin D deficiency and overweight affects muscle mass and function in older post-menopausal women. Aging Clin Exp Res. 2018 Jun;30(6):625-631. doi: 10.1007/s40520-018-0921-1. Epub 2018 Feb 27. PMID: 29488185.

Geoffroy M, Charlot-Lambrecht I, Chrusciel J, Gaubil-Kaladjian I, Diaz-Cives A, Eschard JP, Salmon JH. Impact of Bariatric Surgery on Bone Mineral Density: Observational Study of 110 Patients Followed up in a Specialized Center for the Treatment of Obesity in France. Obes Surg. 2019 Jun;29(6):1765-1772. doi: 10.1007/s11695-019-03719-5.

Reviewer #2: Dear Editor,

in this paper, the authors aimed to investigate the current practice patterns of Korean medicine (KM) treatment for obesity among Korean medical doctors (KMDs) through a questionnaire constructed and distributed to 21,788 KM doctors, consisted of respondent characteristics, state of treated patient, diagnosis, treatment, and usage pattern of herbal medicine for obesity.

Results show that Bioelectric impedance and KM Obesity Pattern Identification Questionnaire are routinely used for diagnosis and herbal medicine is the most commonly used for obesity treatment by KMDs, and Taeeumjowui-tang is the most frequently prescribed. Ephedrae Herba, is identified as the most used herbs for weight loss.

Among the limitations of this study, the success rate of weight loss and the incidence of side effects by KM treatment for obesity are not elicited. The study was based on a self-report survey with a possibility of bias, such as insincere responses, exaggerated treatment effectiveness, and distortion in the number of patients and in the occurrence of adverse events.

The paper has serious flaws and the topic is not very interesting.

In my opinion is not suitable for publication.

6. PLOS authors have the option to publish the peer review history of their article (what does this mean?). If published, this will include your full peer review and any attached files.

Reviewer #1: No

Reviewer #2: No

---

## [Author Response · Author response to Decision Letter 0]

9 Dec 2021

Response to reviewers’ comments. 

Reviewer #1: Dear Authors, the study is interesting and this is an emerging topic. However, there are some issues that should be addressed with a reply point-by-point.

Q1. Which grade of obesity is usually treated with KM? please address with appropriate bibliography.

Response: The authors appreciate reviewer’s helpful comment. Since most of KM treatment for obesity are not covered by the Korean National Health Insurance, basic information on the grades of obesity usually treated with KM has not been collected as health insurance statistics. However, in the present study, 43.3% of KMD respondents (n = 469) reported that the average obesity level of the patients treated was the Obesity Class I (25 ≤ Body mass index (BMI) ≤ 29.9), followed by the Obesity Class II (BMI 30 ~ 34.9, 27.2%, n = 295) and overweight patients (23 ≤ BMI ≤ 24.9, 26.2%, n = 284). Furthermore, Cheon and Jang reported that the proportion of patients taking herbal medicine according to BMI grade based on survey data [1], which indicated that among those taking herbal medicine as a weight control strategy, 51.7% had a BMI of less than 25, 33.2% had a BMI of 25 - 29.9, and 15.1% had a BMI of over 30. Given the result of the present study that almost all responding KMDs prescribe herbal medicines for weight loss, it is considered that KM is usually conducted to overweight and Obesity Class I patients. Even normal weight population are received KM treatment for aesthetic purposes, and although some studies has been conducted on the effect of KM treatment on severe obesity (BMI over 30 kg/m2) [2, 3], additional research is needed on the effects of KM treatment on population group of specific obesity level. We further discussed grade of obesity usually treated by KM in the revised manuscript with appropriate bibliography (Page 12-13, lines 256–273).

Q2. Is the KM first line treatment or alternative treatment? for which obesity grade/s.? Please clarify

Response: The authors appreciate reviewer’s helpful comment. The first line treatment for obesity is lifestyle interventions such as diet control, physical activity, and behavior therapy. If a first line treatment for obesity is unsuccessful, medication and bariatric surgery are considered as secondary therapeutic options. [4]. Although bariatric surgery has been covered by the National Health Insurance since 2019 in Korea, it is only allowed for morbidly obese patients (BMI ≥ 35 kg/m2 or BMI ≥ 30 kg/m2 with comorbidities, such as hypertension and diabetes). In addition, only two anti-obesity drugs, orlistat and lorcaserin, are approved for long-term use in Korea; however, serious side effects, such as liver injury, acute kidney injury, pancreatitis, and cardiac valvulopathy, remain [5]. KM is recognized as the double axis of the Korean health care system along with conventional Western medicine. Supportive evidence has been also accumulated on the safety and effectiveness of herbal medicine for obesity compared with conventional medicines, placebos, or lifestyle control [6, 7]. Moreover, acupuncture is an effective intervention for obesity and its anti-obesity effects may be maximized when combined with lifestyle control [8, 9].

Although the results of the present study showed that KMD treats the patient with Obesity Class I (25 ≤ BMI ≤ 29.9) the most (43.3%, n = 469), and followed by the Obesity Class II (27.2%, n = 295) and overweight patients (26.2%, n = 284), it is needed to be more elucidated obesity level for which KM treatment is most effective. Several studies reported that herbal medicines have a more positive weight loss effect for the patients with higher initial BMI, the more positive the weight loss [2, 10]. Unlike herbal medicine, acupuncture may be more effective for overweight patients than for obese patients [9]. Therefore, acupuncture combined with lifestyle modification could provide a promising therapeutic option to those not requiring pharmacotherapy and bariatric surgery. We further discussed a first line treatment and role of KM treatment for obesity in Korean healthcare system in the revised manuscript (Page 4-5, line 63-75; Page 12-13, lines 256–273).

Q3. Line 262. Obesity class I is >30 kg/m2 in western literature. Please clarify your statement and table; moreover, please provide appropriate bibliography

Response: The authors appreciate reviewer’s critical comment. In western literature, Adult obesity is defined as BMI ≥ 30 kg/m2 and overweight (also known as “pre-obese”) is defined as BMI between 25 and 29.9 kg/m2. However, East Asians generally have higher body fat percentages than non-Asians at the same BMI. Thus, WHO recommended new obesity classification of weight by BMI in adult Asians [11]. The Korean Society for the Study of Obesity also presented new obesity diagnostic criteria in the guidelines for the management of obesity in Korea which based partly on an analysis of data from the Korean National Health Insurance Service Health Checkup database collected from 2006 to 2015 including a total of 84,690,131 Korean adults [4]. The classification of obesity into classes I, II, and III relies on adult BMI, in accordance with WHO guidelines for the Asia-Pacific region [11]. Class I obesity is defined as BMI 25 kg/m2 to less than 30 kg/m2, class II obesity was defined as BMI 30 kg/m2 to less than 35 kg/m2, and class III obesity was newly defined in 2018 as greater than 35 kg/m2 [4]. We clarify the literature basis of the classification of obesity by BMI in Asia-Pacific region was represented in the revised manuscript and table legend, as reviewer’s suggestion (Page 7, lines 124–136, the legend of Table 2).

Q4. Bariatric Surgery is gaining popularity in western medicine for treating obesity. The key factor for such growing popularity is the improvement in obesity-related comorbidities. On the other hand, bariatric surgery could cause malabsorption that leads to several nutritional deficiencies requiring long-term supplementation. Could you find such pros and cons in KM?

Please expand the discussion mentioning the listed studies.

Losco L, Roxo AC, Roxo CW, Lo Torto F, Bolletta A, de Sire A, Aksoyler D, Ribuffo D, Cigna E, Roxo CP. Lower Body Lift After Bariatric Surgery: 323 Consecutive Cases Over 10-Year Experience. Aesthetic Plast Surg. 2020 Apr;44(2):421-432. doi: 10.1007/s00266-019-01543-x.

Gimigliano F, Moretti A, de Sire A, Calafiore D, Iolascon G. The combination of vitamin D deficiency and overweight affects muscle mass and function in older post-menopausal women. Aging Clin Exp Res. 2018 Jun;30(6):625-631. doi: 10.1007/s40520-018-0921-1. Epub 2018 Feb 27. PMID: 29488185.

Geoffroy M, Charlot-Lambrecht I, Chrusciel J, Gaubil-Kaladjian I, Diaz-Cives A, Eschard JP, Salmon JH. Impact of Bariatric Surgery on Bone Mineral Density: Observational Study of 110 Patients Followed up in a Specialized Center for the Treatment of Obesity in France. Obes Surg. 2019 Jun;29(6):1765-1772. doi: 10.1007/s11695-019-03719-5.

Response: The authors appreciate reviewer’s helpful comment. Weight loss is a primary and definitive method for obesity, and various methods for weight control strategies have been used and investigated. Lifestyle intervention, such as diet control and physical activity, are generally recognized as a first line treatment and safe therapy. However, it requires a significant level of commitment from the participants, thereby having limited long-term durability [12]. In Korea, bariatric surgery has been covered by the national health insurance since 2019, and it is very effective for losing weight and improving obesity-related comorbidities [13, 14]. In addition, lower body lift, which restore body contour in postbariatric patients could improve overall quality of life by resulting in pleasant aesthetic outcomes to the obese patients [15]. Despite these advantages, bariatric surgery is only employed to severely obese patients, and postoperative surgical complications due to invasive surgical methods still remain. Moreover, decreased stomach capacity and oral intake, and hormonal changes due to the surgery could cause malabsorption that leads to several nutritional deficiencies requiring long-term supplementation [16]. In particular, in post-menopausal women with obesity, these nutrient deficiencies could increase the risk of skeletal muscles weakness, and metabolic bone disease leading to bone loss and predisposition to fracture [16-18]. Even if weight loss is achieved, its long-term results are generally poor. As a physiological response to weight loss, our body changes various hormones and energy consumption. Hence, control food intake and energy expenditure for maintaining weight loss and preventing weight regain is still challenged after weight reduction. However, sustaining reduced weight can be difficult with lifestyle changes alone, and additional medical therapies may help maintain weight loss. In contrast to bariatric surgery or conventional drugs for obesity, KM, which is represented by treatment using herbal medicine and acupuncture, has the advantage that it can be practiced in parallel with lifestyle modification, and it is associated fewer adverse events. Acupuncture, including not only manual acupuncture but also auricular acupuncture, electroacupuncture, pharmacopuncture, and catgut embedding, effectively treats overweight/obesity by suppressing appetite and relieving hunger and fatigue during losing weight, and its effects on weight loss may be maximized when it combined with lifestyle modification [8, 9]. Herbal medicine including medicinal herbs, their active compounds, and mixed herbal preparations, has beneficial effects on obesity, such as weight loss, waist-hip ratio reduction, body fat reduction, and food intake reduction [7, 19], and is effective for obesity compared with conventional drugs, placebos or lifestyle control with their effectiveness [6]. In addition, it has been reported its pharmacological properties, such as controlling appetite, inhibiting pancreatic lipase activity, stimulating thermogenesis and lipid metabolism, increasing satiety, promoting lipolysis, regulating adipogenesis, and inducing apoptosis in adipocytes [20]. Given these anti-obesity properties of acupuncture and herbal medicine, KM treatment may help to alleviate these adaptive physiological responses after weight loss. KM treatment generally focuses on clinical symptoms and subjective signs, such as overall body conditions, digestive symptoms, pain, and sleeping habit. Thus, herbal medicine and acupuncture may reduce side effects from pharmacotherapy and bariatric surgery. Additional studies are also required to evaluate the efficacy and safety of herbal medicine and acupuncture in combination with other approved conventional drugs for obesity and bariatric surgery.

However, there is currently insufficient evidence to recommend for several herbal medicines weight loss due to low quality or small sample size of clinical trial [6, 21]. Although the safety and effectiveness of KM treatment for obesity is recognized, further large-scale and long-term clinical trials need to be conducted for evaluating KM treatment. In addition, active ingredients in herbs as well as its molecular target and underlying mechanism of action should also be determined. In particular, a direct causal relationship between Ephedae Herba, the herb most frequently prescribed for weight loss by KMDs in this study, and reported adverse events has not been established, thus further safety studies should clarify potential adverse reaction. KM is based on prescribing herbal medicines according to KM syndrome differentiation diagnosis, but only half of the KMDs in the present study used KM syndrome differentiation diagnosis for obesity. Han et al. warned that the administration of herbal medicines as not based on the KM syndrome differentiation diagnosis for obesity may contribute to the occurrence of adverse reactions base on the result of study [7]; thus, more attention should be paid to the use of herbal medicine for weight loss. As reviewer’s suggestion, we further discussed pros and cons of bariatric surgery and pros and KM treatment in the revised manuscript (Page 16-18, lines 362–371, 377-384, 388-413). 

Reviewer #2: Dear Editor, in this paper, the authors aimed to investigate the current practice patterns of Korean medicine (KM) treatment for obesity among Korean medical doctors (KMDs) through a questionnaire constructed and distributed to 21,788 KM doctors, consisted of respondent characteristics, state of treated patient, diagnosis, treatment, and usage pattern of herbal medicine for obesity. Results show that Bioelectric impedance and KM Obesity Pattern Identification Questionnaire are routinely used for diagnosis and herbal medicine is the most commonly used for obesity treatment by KMDs, and Taeeumjowui-tang is the most frequently prescribed. Ephedrae Herba, is identified as the most used herbs for weight loss. Among the limitations of this study, the success rate of weight loss and the incidence of side effects by KM treatment for obesity are not elicited. The study was based on a self-report survey with a possibility of bias, such as insincere responses, exaggerated treatment effectiveness, and distortion in the number of patients and in the occurrence of adverse events. The paper has serious flaws and the topic is not very interesting. In my opinion is not suitable for publication.

Response; We agree with the reviewer that it has the limitation due to the self-report survey. We have mentioned the limitation that the success rate of weight loss and the incidence of side effects by KM treatment for obesity are not elicited in this study. In addition, due to the design of this study based on self-reported survey, deriving the success rate and incidence rate may suggest to inappropriate results. Nevertheless, it has significance that it is the first national survey showing the current clinical practice pattern of KMDs for treating obesity. The overall effectiveness and safety of KM treatment for obesity has been well documented through various study designs, such as retrospective studies, systematic reviews, and meta-analyses [2, 6, 7, 9, 22]. According to the retrospective review of 124 patients who had taken Gamitaeeumjowee-tang for 10 weeks, the overall rate of adverse events was 37.1% during Week 2 - 4 and 16.9% at Week 10 (for causality of adverse events using the WHO-Uppsala Monitoring Centre causality categories, 52.2% were evaluated as “possible” at Week 2-4 and 57.1% were evaluated as “unlikely” at Week 10) [23]. We soon plan to build prospective registry of herbal medicine for weight loss to register herbal medicine, and any side effects, including their causality and severity. This will enable the acute statistical investigation of the occurrence of side effects (Page 18-19, lines 422–437). 

References

1. Cheon C, Jang BH. Trends for weight control strategies in Korean adults using the Korea national health and nutrition examination survey from 2007 to 2017. Explore (NY). 2020. doi: 10.1016/j.explore.2020.03.010 PMID: 32434671

2. Lee Y-H, Go N-G, Min D-L. Retrospective study about the effectiveness of Korean medicine treatment on 254 patients visited obesity clinic. Journal of Korean Medicine for Obesity Research. 2015;15(01):33-7. doi: 10.15429/jkomor.2015.15.1.33 

3. Kim S, Han K, Kwon O, Lee W, Yoon C, Lee J-H. Effect of Korean medicine treatment including Korean medicine counselling on weight loss in patients with morbid obesity: A retrospective chart review. Journal of Korean Medicine for Obesity Research. 2021;21(1):22-31. doi: 10.15429/jkomor.2021.21.1.22 

4. Seo MH, Lee WY, Kim SS, Kang JH, Kang JH, Kim KK, et al. 2018 Korean society for the study of obesity guideline for the management of obesity in Korea. J Obes Metab Syndr. 2019;28(1):40-5. doi: 10.7570/jomes.2019.28.1.40 PMID: 31089578

5. Kim KK. Safety of anti-obesity drugs approved for long-term use. The Korean Journal of Obesity. 2015;24(1):17-27. doi: 10.7570/kjo.2015.24.1.17 

6. Park JH, Lee MJ, Song MY, Bose S, Shin BC, Kim HJ. Efficacy and safety of mixed oriental herbal medicines for treating human obesity: A systematic review of randomized clinical trials. J Med Food. 2012;15(7):589-97. doi: 10.1089/jmf.2011.1982 PMID: 22612295

7. Han K, Lee M-J, Kim H. Systematic review on herbal treatment for obesity in adults. Journal of Korean Medicine Rehabilitation. 2016;26(4):23-35. doi: 10.18325/jkmr.2016.26.4.23 

8. Fang S, Wang M, Zheng Y, Zhou S, Ji G. Acupuncture and lifestyle modification treatment for obesity: A meta-analysis. Am J Chin Med. 2017;45(2):239-54. doi: 10.1142/S0192415X1750015X PMID: 28231746

9. Kim SY, Shin IS, Park YJ. Effect of acupuncture and intervention types on weight loss: A systematic review and meta-analysis. Obes Rev. 2018;19(11):1585-96. doi: 10.1111/obr.12747 PMID: 30180304

10. Lee S-J, Kim W-I. A clinical study about the effects of oriental medical therapy on obesity and different effects between groups. The Journal of Korean Oriental Medical Ophthalmology and Otolaryngology and Dermatology. 2012;25(3):97-112. doi: 10.6114/jkood.2012.25.3.097 

11. World Health Organization. Regional Office for the Western P. The asia-pacific perspective : Redefining obesity and its treatment: Sydney : Health Communications Australia; 2000 2000.

12. Sarlio-Lähteenkorva S. ‘The battle is not over after weight loss’: Stories of successful weight loss maintenance. Health:. 2000;4(1):73-88. 

13. Kim BY, Kang SM, Kang JH, Kang SY, Kim KK, Kim KB, et al. 2020 Korean society for the study of obesity guidelines for the management of obesity in korea. J Obes Metab Syndr. 2021;30(2):81-92. doi: 10.7570/jomes21022 PMID: 34045368

14. Park JY, Heo Y, Kim YJ, Park JM, Kim SM, Park DJ, et al. Long-term effect of bariatric surgery versus conventional therapy in obese Korean patients: A multicenter retrospective cohort study. Ann Surg Treat Res. 2019;96(6):283-9. doi: 10.4174/astr.2019.96.6.283 PMID: 31183332

15. Losco L, Roxo AC, Roxo CW, Lo Torto F, Bolletta A, de Sire A, et al. Lower body lift after bariatric surgery: 323 consecutive cases over 10-year experience. Aesthetic Plast Surg. 2020;44(2):421-32. doi: 10.1007/s00266-019-01543-x PMID: 31748908

16. Feng XC, Burch M. Management of postoperative complications following bariatric and metabolic procedures. Surg Clin North Am. 2021;101(5):731-53. doi: 10.1016/j.suc.2021.05.017 PMID: 34537140

17. Geoffroy M, Charlot-Lambrecht I, Chrusciel J, Gaubil-Kaladjian I, Diaz-Cives A, Eschard JP, et al. Impact of bariatric surgery on bone mineral density: Observational study of 110 patients followed up in a specialized center for the treatment of obesity in france. Obes Surg. 2019;29(6):1765-72. doi: 10.1007/s11695-019-03719-5 PMID: 30734230

18. Gimigliano F, Moretti A, de Sire A, Calafiore D, Iolascon G. The combination of vitamin D deficiency and overweight affects muscle mass and function in older post-menopausal women. Aging Clin Exp Res. 2018;30(6):625-31. doi: 10.1007/s40520-018-0921-1 PMID: 29488185

19. Hasani-Ranjbar S, Nayebi N, Larijani B, Abdollahi M. A systematic review of the efficacy and safety of herbal medicines used in the treatment of obesity. World J Gastroenterol. 2009;15(25):3073-85. doi: 10.3748/wjg.15.3073 PMID: 19575486

20. Saad B, Ghareeb B, Kmail A. Metabolic and epigenetics action mechanisms of antiobesity medicinal plants and phytochemicals. Evid Based Complement Alternat Med. 2021;2021:9995903. doi: 10.1155/2021/9995903 PMID: 34211580

21. Maunder A, Bessell E, Lauche R, Adams J, Sainsbury A, Fuller NR. Effectiveness of herbal medicines for weight loss: A systematic review and meta-analysis of randomized controlled trials. Diabetes Obes Metab. 2020;22(6):891-903. doi: 10.1111/dom.13973 PMID: 31984610

22. Sui Y, Zhao HL, Wong VC, Brown N, Li XL, Kwan AK, et al. A systematic review on use of chinese medicine and acupuncture for treatment of obesity. Obes Rev. 2012;13(5):409-30. doi: 10.1111/j.1467-789X.2011.00979.x PMID: 22292480

23. Yoon N-R, Yoo Y-J, Kim M-j, Kim S-Y, Lim Y-W, Lim HH, et al. Analysis of adverse events in weight loss program in combination with Gamitaeeumjowee-tang and low-calorie diet. Journal of Korean Medicine for Obesity Research. 2018;18(1):1-9. doi: 10.15429/jkomor.2018.18.1.1

---

## [Editor Report · Decision Letter 1]

16 Feb 2022

PONE-D-21-26587R1A national survey on current clinical practice pattern of Korean Medicine doctors for treating obesityPLOS ONE

Dear Dr. Kim,

Thank you for submitting your manuscript to PLOS ONE. After careful consideration, we feel that it has merit but does not fully meet PLOS ONE’s publication criteria as it currently stands. Therefore, we invite you to submit a revised version of the manuscript that addresses the points raised during the review process.

We look forward to receiving your revised manuscript.

Kind regards,

Alessandro de Sire, M.D.

Academic Editor

PLOS ONE
---

## [Author Response · Author response to Decision Letter 1]

21 Feb 2022

Response to reviewers’ comments. 

Reviewer #1: Dear Authors, the study is interesting and this is an emerging topic. However, there are some issues that should be addressed with a reply point-by-point.

Q1. Which grade of obesity is usually treated with KM? please address with appropriate bibliography.

Response: The authors appreciate reviewer’s helpful comment. Since most of KM treatment for obesity are not covered by the Korean National Health Insurance, basic information on the grades of obesity usually treated with KM has not been collected as health insurance statistics. However, in the present study, 43.3% of KMD respondents (n = 469) reported that the average obesity level of the patients treated was the Obesity Class I (25 ≤ Body mass index (BMI) ≤ 29.9), followed by the Obesity Class II (BMI 30 ~ 34.9, 27.2%, n = 295) and overweight patients (23 ≤ BMI ≤ 24.9, 26.2%, n = 284). Furthermore, Cheon and Jang reported that the proportion of patients taking herbal medicine according to BMI grade based on survey data [1], which indicated that among those taking herbal medicine as a weight control strategy, 51.7% had a BMI of less than 25, 33.2% had a BMI of 25 - 29.9, and 15.1% had a BMI of over 30. Given the result of the present study that almost all responding KMDs prescribe herbal medicines for weight loss, it is considered that KM is usually conducted to overweight and Obesity Class I patients. Even normal weight population are received KM treatment for aesthetic purposes, and although some studies has been conducted on the effect of KM treatment on severe obesity (BMI over 30 kg/m2) [2, 3], additional research is needed on the effects of KM treatment on population group of specific obesity level. We further discussed grade of obesity usually treated by KM in the revised manuscript with appropriate bibliography (Page 12-13, lines 256–273).

Q2. Is the KM first line treatment or alternative treatment? for which obesity grade/s.? Please clarify

Response: The authors appreciate reviewer’s helpful comment. The first line treatment for obesity is lifestyle interventions such as diet control, physical activity, and behavior therapy. If a first line treatment for obesity is unsuccessful, medication and bariatric surgery are considered as secondary therapeutic options. [4]. Although bariatric surgery has been covered by the National Health Insurance since 2019 in Korea, it is only allowed for morbidly obese patients (BMI ≥ 35 kg/m2 or BMI ≥ 30 kg/m2 with comorbidities, such as hypertension and diabetes). In addition, only two anti-obesity drugs, orlistat and lorcaserin, are approved for long-term use in Korea; however, serious side effects, such as liver injury, acute kidney injury, pancreatitis, and cardiac valvulopathy, remain [5]. KM is recognized as the double axis of the Korean health care system along with conventional Western medicine. Supportive evidence has been also accumulated on the safety and effectiveness of herbal medicine for obesity compared with conventional medicines, placebos, or lifestyle control [6, 7]. Moreover, acupuncture is an effective intervention for obesity and its anti-obesity effects may be maximized when combined with lifestyle control [8, 9].

Although the results of the present study showed that KMD treats the patient with Obesity Class I (25 ≤ BMI ≤ 29.9) the most (43.3%, n = 469), and followed by the Obesity Class II (27.2%, n = 295) and overweight patients (26.2%, n = 284), it is needed to be more elucidated obesity level for which KM treatment is most effective. Several studies reported that herbal medicines have a more positive weight loss effect for the patients with higher initial BMI, the more positive the weight loss [2, 10]. Unlike herbal medicine, acupuncture may be more effective for overweight patients than for obese patients [9]. Therefore, acupuncture combined with lifestyle modification could provide a promising therapeutic option to those not requiring pharmacotherapy and bariatric surgery. We further discussed a first line treatment and role of KM treatment for obesity in Korean healthcare system in the revised manuscript (Page 4-5, line 63-75; Page 12-13, lines 256–273).

Q3. Line 262. Obesity class I is >30 kg/m2 in western literature. Please clarify your statement and table; moreover, please provide appropriate bibliography

Response: The authors appreciate reviewer’s critical comment. In western literature, Adult obesity is defined as BMI ≥ 30 kg/m2 and overweight (also known as “pre-obese”) is defined as BMI between 25 and 29.9 kg/m2. However, East Asians generally have higher body fat percentages than non-Asians at the same BMI. Thus, WHO recommended new obesity classification of weight by BMI in adult Asians [11]. The Korean Society for the Study of Obesity also presented new obesity diagnostic criteria in the guidelines for the management of obesity in Korea which based partly on an analysis of data from the Korean National Health Insurance Service Health Checkup database collected from 2006 to 2015 including a total of 84,690,131 Korean adults [4]. The classification of obesity into classes I, II, and III relies on adult BMI, in accordance with WHO guidelines for the Asia-Pacific region [11]. Class I obesity is defined as BMI 25 kg/m2 to less than 30 kg/m2, class II obesity was defined as BMI 30 kg/m2 to less than 35 kg/m2, and class III obesity was newly defined in 2018 as greater than 35 kg/m2 [4]. We clarify the literature basis of the classification of obesity by BMI in Asia-Pacific region was represented in the revised manuscript and table legend, as reviewer’s suggestion (Page 7, lines 124–136, the legend of Table 2).

Q4. Bariatric Surgery is gaining popularity in western medicine for treating obesity. The key factor for such growing popularity is the improvement in obesity-related comorbidities. On the other hand, bariatric surgery could cause malabsorption that leads to several nutritional deficiencies requiring long-term supplementation. Could you find such pros and cons in KM?

Please expand the discussion mentioning the listed studies.

Losco L, Roxo AC, Roxo CW, Lo Torto F, Bolletta A, de Sire A, Aksoyler D, Ribuffo D, Cigna E, Roxo CP. Lower Body Lift After Bariatric Surgery: 323 Consecutive Cases Over 10-Year Experience. Aesthetic Plast Surg. 2020 Apr;44(2):421-432. doi: 10.1007/s00266-019-01543-x.

Gimigliano F, Moretti A, de Sire A, Calafiore D, Iolascon G. The combination of vitamin D deficiency and overweight affects muscle mass and function in older post-menopausal women. Aging Clin Exp Res. 2018 Jun;30(6):625-631. doi: 10.1007/s40520-018-0921-1. Epub 2018 Feb 27. PMID: 29488185.

Geoffroy M, Charlot-Lambrecht I, Chrusciel J, Gaubil-Kaladjian I, Diaz-Cives A, Eschard JP, Salmon JH. Impact of Bariatric Surgery on Bone Mineral Density: Observational Study of 110 Patients Followed up in a Specialized Center for the Treatment of Obesity in France. Obes Surg. 2019 Jun;29(6):1765-1772. doi: 10.1007/s11695-019-03719-5.

Response: The authors appreciate reviewer’s helpful comment. Weight loss is a primary and definitive method for obesity, and various methods for weight control strategies have been used and investigated. Lifestyle intervention, such as diet control and physical activity, are generally recognized as a first line treatment and safe therapy. However, it requires a significant level of commitment from the participants, thereby having limited long-term durability [12]. In Korea, bariatric surgery has been covered by the national health insurance since 2019, and it is very effective for losing weight and improving obesity-related comorbidities [13, 14]. In addition, lower body lift, which restore body contour in postbariatric patients could improve overall quality of life by resulting in pleasant aesthetic outcomes to the obese patients [15]. Despite these advantages, bariatric surgery is only employed to severely obese patients, and postoperative surgical complications due to invasive surgical methods still remain. Moreover, decreased stomach capacity and oral intake, and hormonal changes due to the surgery could cause malabsorption that leads to several nutritional deficiencies requiring long-term supplementation [16]. In particular, in post-menopausal women with obesity, these nutrient deficiencies could increase the risk of skeletal muscles weakness, and metabolic bone disease leading to bone loss and predisposition to fracture [16-18]. Even if weight loss is achieved, its long-term results are generally poor. As a physiological response to weight loss, our body changes various hormones and energy consumption. Hence, control food intake and energy expenditure for maintaining weight loss and preventing weight regain is still challenged after weight reduction. However, sustaining reduced weight can be difficult with lifestyle changes alone, and additional medical therapies may help maintain weight loss. In contrast to bariatric surgery or conventional drugs for obesity, KM, which is represented by treatment using herbal medicine and acupuncture, has the advantage that it can be practiced in parallel with lifestyle modification, and it is associated fewer adverse events. Acupuncture, including not only manual acupuncture but also auricular acupuncture, electroacupuncture, pharmacopuncture, and catgut embedding, effectively treats overweight/obesity by suppressing appetite and relieving hunger and fatigue during losing weight, and its effects on weight loss may be maximized when it combined with lifestyle modification [8, 9]. Herbal medicine including medicinal herbs, their active compounds, and mixed herbal preparations, has beneficial effects on obesity, such as weight loss, waist-hip ratio reduction, body fat reduction, and food intake reduction [7, 19], and is effective for obesity compared with conventional drugs, placebos or lifestyle control with their effectiveness [6]. In addition, it has been reported its pharmacological properties, such as controlling appetite, inhibiting pancreatic lipase activity, stimulating thermogenesis and lipid metabolism, increasing satiety, promoting lipolysis, regulating adipogenesis, and inducing apoptosis in adipocytes [20]. Given these anti-obesity properties of acupuncture and herbal medicine, KM treatment may help to alleviate these adaptive physiological responses after weight loss. KM treatment generally focuses on clinical symptoms and subjective signs, such as overall body conditions, digestive symptoms, pain, and sleeping habit. Thus, herbal medicine and acupuncture may reduce side effects from pharmacotherapy and bariatric surgery. Additional studies are also required to evaluate the efficacy and safety of herbal medicine and acupuncture in combination with other approved conventional drugs for obesity and bariatric surgery.

However, there is currently insufficient evidence to recommend for several herbal medicines weight loss due to low quality or small sample size of clinical trial [6, 21]. Although the safety and effectiveness of KM treatment for obesity is recognized, further large-scale and long-term clinical trials need to be conducted for evaluating KM treatment. In addition, active ingredients in herbs as well as its molecular target and underlying mechanism of action should also be determined. In particular, a direct causal relationship between Ephedae Herba, the herb most frequently prescribed for weight loss by KMDs in this study, and reported adverse events has not been established, thus further safety studies should clarify potential adverse reaction. KM is based on prescribing herbal medicines according to KM syndrome differentiation diagnosis, but only half of the KMDs in the present study used KM syndrome differentiation diagnosis for obesity. Han et al. warned that the administration of herbal medicines as not based on the KM syndrome differentiation diagnosis for obesity may contribute to the occurrence of adverse reactions base on the result of study [7]; thus, more attention should be paid to the use of herbal medicine for weight loss. As reviewer’s suggestion, we further discussed pros and cons of bariatric surgery and pros and KM treatment in the revised manuscript (Page 16-18, lines 362–371, 377-384, 388-413). 

Reviewer #2: Dear Editor, in this paper, the authors aimed to investigate the current practice patterns of Korean medicine (KM) treatment for obesity among Korean medical doctors (KMDs) through a questionnaire constructed and distributed to 21,788 KM doctors, consisted of respondent characteristics, state of treated patient, diagnosis, treatment, and usage pattern of herbal medicine for obesity. Results show that Bioelectric impedance and KM Obesity Pattern Identification Questionnaire are routinely used for diagnosis and herbal medicine is the most commonly used for obesity treatment by KMDs, and Taeeumjowui-tang is the most frequently prescribed. Ephedrae Herba, is identified as the most used herbs for weight loss. Among the limitations of this study, the success rate of weight loss and the incidence of side effects by KM treatment for obesity are not elicited. The study was based on a self-report survey with a possibility of bias, such as insincere responses, exaggerated treatment effectiveness, and distortion in the number of patients and in the occurrence of adverse events. The paper has serious flaws and the topic is not very interesting. In my opinion is not suitable for publication.

Response; We agree with the reviewer that it has the limitation due to the self-report survey. We have mentioned the limitation that the success rate of weight loss and the incidence of side effects by KM treatment for obesity are not elicited in this study. In addition, due to the design of this study based on self-reported survey, deriving the success rate and incidence rate may suggest to inappropriate results. Nevertheless, it has significance that it is the first national survey showing the current clinical practice pattern of KMDs for treating obesity. The overall effectiveness and safety of KM treatment for obesity has been well documented through various study designs, such as retrospective studies, systematic reviews, and meta-analyses [2, 6, 7, 9, 22]. According to the retrospective review of 124 patients who had taken Gamitaeeumjowee-tang for 10 weeks, the overall rate of adverse events was 37.1% during Week 2 - 4 and 16.9% at Week 10 (for causality of adverse events using the WHO-Uppsala Monitoring Centre causality categories, 52.2% were evaluated as “possible” at Week 2-4 and 57.1% were evaluated as “unlikely” at Week 10) [23]. We soon plan to build prospective registry of herbal medicine for weight loss to register herbal medicine, and any side effects, including their causality and severity. This will enable the acute statistical investigation of the occurrence of side effects (Page 18-19, lines 422–437). 

 

References

1. Cheon C, Jang BH. Trends for weight control strategies in Korean adults using the Korea national health and nutrition examination survey from 2007 to 2017. Explore (NY). 2020. doi: 10.1016/j.explore.2020.03.010 PMID: 32434671

2. Lee Y-H, Go N-G, Min D-L. Retrospective study about the effectiveness of Korean medicine treatment on 254 patients visited obesity clinic. Journal of Korean Medicine for Obesity Research. 2015;15(01):33-7. doi: 10.15429/jkomor.2015.15.1.33 

3. Kim S, Han K, Kwon O, Lee W, Yoon C, Lee J-H. Effect of Korean medicine treatment including Korean medicine counselling on weight loss in patients with morbid obesity: A retrospective chart review. Journal of Korean Medicine for Obesity Research. 2021;21(1):22-31. doi: 10.15429/jkomor.2021.21.1.22 

4. Seo MH, Lee WY, Kim SS, Kang JH, Kang JH, Kim KK, et al. 2018 Korean society for the study of obesity guideline for the management of obesity in Korea. J Obes Metab Syndr. 2019;28(1):40-5. doi: 10.7570/jomes.2019.28.1.40 PMID: 31089578

5. Kim KK. Safety of anti-obesity drugs approved for long-term use. The Korean Journal of Obesity. 2015;24(1):17-27. doi: 10.7570/kjo.2015.24.1.17 

6. Park JH, Lee MJ, Song MY, Bose S, Shin BC, Kim HJ. Efficacy and safety of mixed oriental herbal medicines for treating human obesity: A systematic review of randomized clinical trials. J Med Food. 2012;15(7):589-97. doi: 10.1089/jmf.2011.1982 PMID: 22612295

7. Han K, Lee M-J, Kim H. Systematic review on herbal treatment for obesity in adults. Journal of Korean Medicine Rehabilitation. 2016;26(4):23-35. doi: 10.18325/jkmr.2016.26.4.23 

8. Fang S, Wang M, Zheng Y, Zhou S, Ji G. Acupuncture and lifestyle modification treatment for obesity: A meta-analysis. Am J Chin Med. 2017;45(2):239-54. doi: 10.1142/S0192415X1750015X PMID: 28231746

9. Kim SY, Shin IS, Park YJ. Effect of acupuncture and intervention types on weight loss: A systematic review and meta-analysis. Obes Rev. 2018;19(11):1585-96. doi: 10.1111/obr.12747 PMID: 30180304

10. Lee S-J, Kim W-I. A clinical study about the effects of oriental medical therapy on obesity and different effects between groups. The Journal of Korean Oriental Medical Ophthalmology and Otolaryngology and Dermatology. 2012;25(3):97-112. doi: 10.6114/jkood.2012.25.3.097 

11. World Health Organization. Regional Office for the Western P. The asia-pacific perspective : Redefining obesity and its treatment: Sydney : Health Communications Australia; 2000 2000.

12. Sarlio-Lähteenkorva S. ‘The battle is not over after weight loss’: Stories of successful weight loss maintenance. Health:. 2000;4(1):73-88. 

13. Kim BY, Kang SM, Kang JH, Kang SY, Kim KK, Kim KB, et al. 2020 Korean society for the study of obesity guidelines for the management of obesity in korea. J Obes Metab Syndr. 2021;30(2):81-92. doi: 10.7570/jomes21022 PMID: 34045368

14. Park JY, Heo Y, Kim YJ, Park JM, Kim SM, Park DJ, et al. Long-term effect of bariatric surgery versus conventional therapy in obese Korean patients: A multicenter retrospective cohort study. Ann Surg Treat Res. 2019;96(6):283-9. doi: 10.4174/astr.2019.96.6.283 PMID: 31183332

15. Losco L, Roxo AC, Roxo CW, Lo Torto F, Bolletta A, de Sire A, et al. Lower body lift after bariatric surgery: 323 consecutive cases over 10-year experience. Aesthetic Plast Surg. 2020;44(2):421-32. doi: 10.1007/s00266-019-01543-x PMID: 31748908

16. Feng XC, Burch M. Management of postoperative complications following bariatric and metabolic procedures. Surg Clin North Am. 2021;101(5):731-53. doi: 10.1016/j.suc.2021.05.017 PMID: 34537140

17. Geoffroy M, Charlot-Lambrecht I, Chrusciel J, Gaubil-Kaladjian I, Diaz-Cives A, Eschard JP, et al. Impact of bariatric surgery on bone mineral density: Observational study of 110 patients followed up in a specialized center for the treatment of obesity in france. Obes Surg. 2019;29(6):1765-72. doi: 10.1007/s11695-019-03719-5 PMID: 30734230

18. Gimigliano F, Moretti A, de Sire A, Calafiore D, Iolascon G. The combination of vitamin D deficiency and overweight affects muscle mass and function in older post-menopausal women. Aging Clin Exp Res. 2018;30(6):625-31. doi: 10.1007/s40520-018-0921-1 PMID: 29488185

19. Hasani-Ranjbar S, Nayebi N, Larijani B, Abdollahi M. A systematic review of the efficacy and safety of herbal medicines used in the treatment of obesity. World J Gastroenterol. 2009;15(25):3073-85. doi: 10.3748/wjg.15.3073 PMID: 19575486

20. Saad B, Ghareeb B, Kmail A. Metabolic and epigenetics action mechanisms of antiobesity medicinal plants and phytochemicals. Evid Based Complement Alternat Med. 2021;2021:9995903. doi: 10.1155/2021/9995903 PMID: 34211580

21. Maunder A, Bessell E, Lauche R, Adams J, Sainsbury A, Fuller NR. Effectiveness of herbal medicines for weight loss: A systematic review and meta-analysis of randomized controlled trials. Diabetes Obes Metab. 2020;22(6):891-903. doi: 10.1111/dom.13973 PMID: 31984610

22. Sui Y, Zhao HL, Wong VC, Brown N, Li XL, Kwan AK, et al. A systematic review on use of chinese medicine and acupuncture for treatment of obesity. Obes Rev. 2012;13(5):409-30. doi: 10.1111/j.1467-789X.2011.00979.x PMID: 22292480

23. Yoon N-R, Yoo Y-J, Kim M-j, Kim S-Y, Lim Y-W, Lim HH, et al. Analysis of adverse events in weight loss program in combination with Gamitaeeumjowee-tang and low-calorie diet. Journal of Korean Medicine for Obesity Research. 2018;18(1):1-9. doi: 10.15429/jkomor.2018.18.1.1

---

## [Editor Report · Decision Letter 2]

14 Mar 2022

A national survey on current clinical practice pattern of Korean Medicine doctors for treating obesity

PONE-D-21-26587R2

Dear Dr. Kim,

We’re pleased to inform you that your manuscript has been judged scientifically suitable for publication and will be formally accepted for publication once it meets all outstanding technical requirements.

Kind regards,

Alessandro de Sire, M.D.

Academic Editor

PLOS ONE

Additional Editor Comments (optional):

The paper could be accepted in this form.
---

## [Editor Report · Acceptance letter]

15 Mar 2022

PONE-D-21-26587R2 

A national survey on current clinical practice pattern of Korean Medicine doctors for treating obesity 

Dear Dr. Kim:

I'm pleased to inform you that your manuscript has been deemed suitable for publication in PLOS ONE. Congratulations! Your manuscript is now with our production department. 

Kind regards, 

on behalf of

Prof. Alessandro de Sire 

Academic Editor

PLOS ONE